# The Innovative Design of the Fire-Fighting Adapter for Forest Machinery

**Richard Hnilica [1,*], Miroslava Ťavodová [1], Michaela Hnilicová [1], Jaroslav Matej [2] and Valéria Messingerová [3]**

1   Department of Manufacturing Technology and Quality Management, Faculty of Technology,
    Technical University in Zvolen, T.G. Masaryka 24, Zvolen 96001, Slovakia; tavodova@tuzvo.sk (M.Ť.);
    michaela.hnilicova@gmail.com (M.H.)
2   Department of Mechanics, Mechanical Engineering and Design, Faculty of Technology,
    Technical University in Zvolen, T.G. Masaryka 24, Zvolen 96001, Slovakia; jaroslav.matej@tuzvo.sk
3   Department of Forest Harvesting, Logistics and Ameliorations, Faculty of Forestry,
    Technical University in Zvolen, T.G. Masaryka 24, Zvolen 96001, Slovakia; messingerova@tuzvo.sk
*   Correspondence: hnilica@tuzvo.sk

**Abstract:** The presented article presents an innovative solution of a fire-fighting adapter based on the basic hypothesis: to provide sufficient technical support in difficult terrain conditions for water transport logistics in order to quickly prevent the spread and destruction of forest fire. At present, when forest fires begin, it is often quite complicated to provide sufficient technical support for the quick prevention and elimination of fires. This fact is largely eliminated by the designed fire-fighting adapter. The mentioned fire-fighting adapter can be used as a fire-fighting mobile device with a base machine of a forest wheeled skidder (LKT), part of the long-distance transport of water in difficult terrain (lake system), a water tank in difficult terrain with the possibility of filling the Bambi bag with a helicopter, part of the long-distance transport of water in the case of a difficult water source without a forest access road network, and a mobile device for emergency transportation of materials in difficult terrain. In addition to the use for fighting forest fires, the fire adapter can also be used to provide for transport of water for forest nurseries (irrigation), freshly planted areas in the event of prolonged drought, the filling of watering-places for forest animals and filling puddles in the dry season. When designing the supporting frame, we used and imitated the evolutionary approach of nature in the form of generative design. The presented paper deals with the use of modern composite materials in the design of superstructures for base machines, which gradually acquire meaning. The main reason for the increasing use of fiberglass is to achieve higher strength and safer weight reduction. This adapter is designed for forest wheel tractors that reach 40% slope availability, are able to work on the stand area, overcome obstacles and are available in sufficient quantities in all Slovak forest areas.

**Keywords:** forest fires; adapter; forest wheeled skidder; design; composite

## 1. Introduction

With regard to forest protection, forest fires are the most drastic form of forest destruction. They have an enormous impact on all forest ecosystems, especially the plants and animals living there. From a technical point of view, a forest fire is a sudden, partly or very uncontrolled extraordinary event limited in time and place. This fact has a negative impact on all social functions of forests (productive and non-productive). It is a complex physical and chemical effect, which are based on non-stationary processes (changing in place and time) burning, gas exchange and heat transfer.

Elimination of forest fires on any scale combines several phenomena. Firstly, sufficient forces and resources for its localization and liquidation. Secondly, the availability of the site and sufficient extinguishing agent (water) in connection with the availability of water resources themselves. Not one area of this vicious circle cannot be prepared in advance. However, making the forest accessible with the preservation, revitalization and construction of the forest road network and water resources should be part of the construction of the forest. All this in accordance with the preservation of production and increasingly presented non-production functions of the forest. [1]

It is necessary to look at safety construction from the point of view that it is not just a question of designing and dimensioning machinery, but the creation of a comprehensive system of filling defined objectives. This means that for adapters, which are used as attachments to forestry machinery, it is necessary to approach the design from two perspectives. The first is the purpose for which the adapters are designed (operational characteristics) and the second is the impact on the operator or other employees working nearby. The task of designers who deal with technical (operational) parameters is to ensure the highest possible efficiency of the machine with the adapter. The role of engineers who deal with security, is to minimize the hazards arising throughout the life of the machine [2,3].

Design and development of pumping appliances (CAS) for fire-fighting in inaccessible forest fire areas are subject to high costs from the point of view of development, and also have limited possibilities of mass production (piece production). The last question is utilization of such a one-purpose machine, mainly the number of uses in relation to the working life of the machine. This is the basic problem which invites alternative solutions. In this paper the solution is based on analysis of the terrain in which the machine operates; the selection of a machines that exist in the present, and is suitable for this environment; the design of a body-adapter, based on a water tank; the selection of suitable materials from the point of view of strength and low weight. Another criterion is the shape of the water tank with emphasis on conservation of the native travel abilities of the base vehicle.

## 2. Materials and Methods

The need to transport water to the nearest intervening fire brigade has become a basic premise in consideration of the suitable design of fire equipment. Based on this hypothesis, the most appropriate alternative is the treatment and use of forest machinery. The starting point for the design will be the definition of the body carrier with maximum provision for slope accessibility and workability in forest terrain. Another important criterion will be the use of available technology (base machine) used in forestry, without the need to intervene in its construction, and eventually the development of new technology. When accepting the basic requirements in relation to workability in forest terrain, forest wheeled skidders (LKT) came into consideration as base machines. These base machines are primarily designed and intended to work in difficult forest terrain.

The main task of an LKT is to move wood from the forest to the place of its further manipulation. In the case of transport of an extinguishing agent, it is a change of the basic working algorithm. The space remains and only the direction of movement changes. For this reason, when considering how to use existing LKTs for these purposes, the design of the fire-fighting adapter will be based on the following basic concepts of its use:

- The use of base machine (LKT) in the form of a pulling device to increase the pumping appliances (CAS)'s accessibility—concept I (Figure 1a),
- The use of the base machine (LKT) as a pulling device for pulling a one-axle or two-axle trailer—concept II (Figure 1b),
- The use of the base machine (LKT) as a carrier for the transport of water (a special purpose machine)—concept III (Figure 1c),
- The use of base machine (LKT) as a carrier for a removable water transport body—concept IV (Figure 1d).

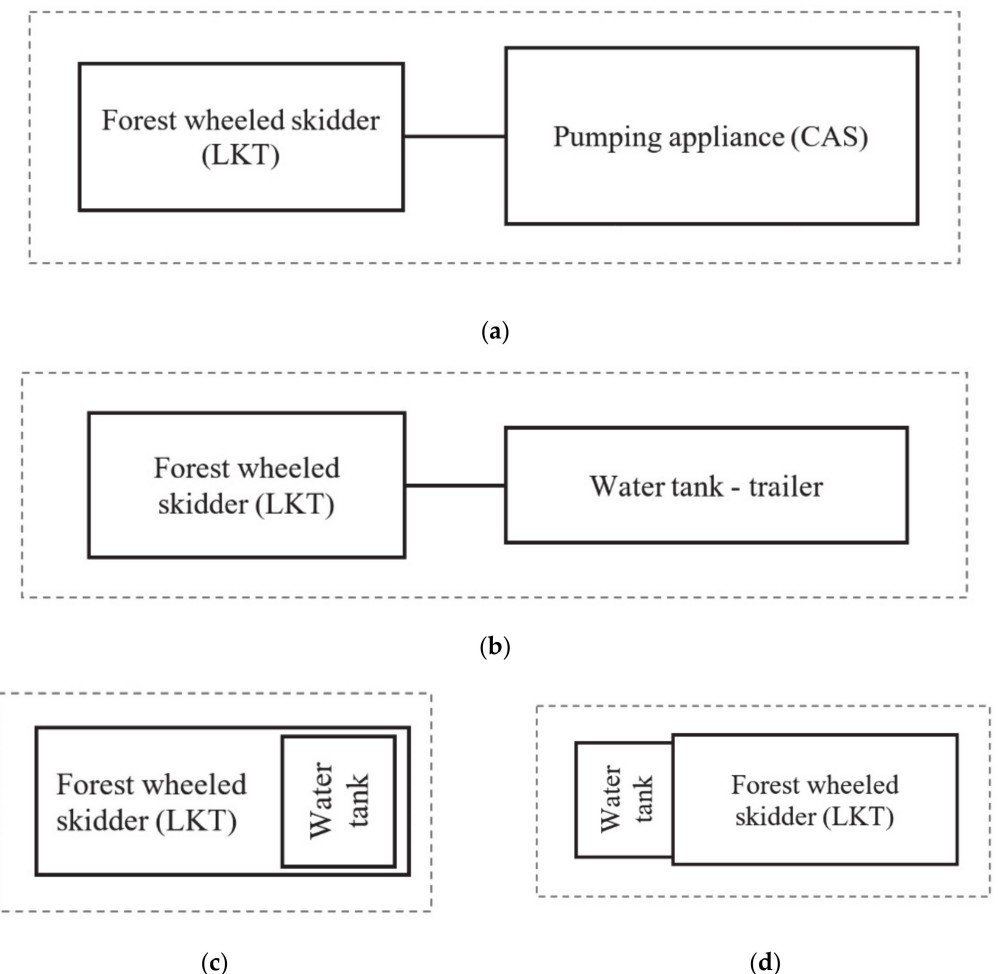

**Figure 1.** The basic concept of using a base machine (forest wheeled skidders (LKT)), (**a**) in the form of traction gear to increase the endurance distance of pumping appliances (CAS); (**b**) in the form of traction gear to increase a one-axle or two-axle trailer; (**c**) as a carrier of a fixed body for water transport—a special purpose machine; (**d**) as a carrier of a detachable body for water transport.

As it can be seen from concept I, the LKT is used as a traction device to increase the uptake of fire-fighting equipment in forest terrain. In this case, it could be pulled by an LKT, e.g., Mercedes-Benz UNIMOG CAS. However, this solution is at the limit of the possibility of this technology with the risk of its damage.

Concept II is subject to the limited capability of the trailer, in relation to its stability when crossing the terrain (encountering difficulties such as churning up the road, the tracks on the road, the stones or root stones, and the tires).

The concept III shows the uniqueness of such a machine, which significantly increases the costs of its production and operation. Small-scale or piece production does not ensure its presence in the area, in the event of a forest fire.

Under concept IV, the original purpose of the LKT is preserved, with minimal constraints on driving characteristics. With a volume of carried fire-extinguishing material, approx. 2000 L, replacing vehicles mounted onto a Praga V3S or Mercedes-Benz UNIMOG chassis. In this way, the original LKT function is retained with minimal delay when applying the body to the rear LKT shield.

Concept I is simple in its design, but for use in mountain forests fires there are the limits to this technique, with the risk of damage to the towed fire-fighting equipment. Concept II, due to its designs, also experiences limiting factors of stability and maneuverability. Concept III is suitable for the purpose of forest fires, but on the other hand it significantly interferes with the construction of the base LKT

machine and changes it from a multi-purpose machine to a single-purpose machine. When designing an alternative location adapter (body), the design was based on two options. Design of a special platform for the rear axle of the LKT, or an adapter for the rear LKT shield. After considering all the pros and cons, the adapter was developed, using the rear LKT shield (Figure 2).

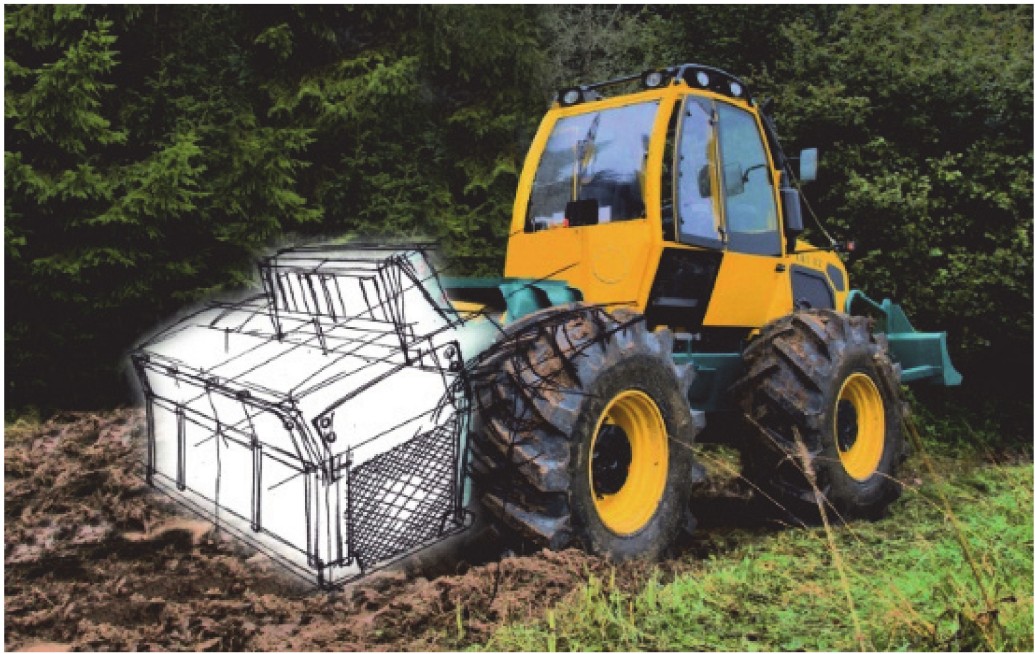

**Figure 2.** The use of a base machine in fires in the forest, as a carrier of a detachable body.

Basic material analyses were performed on:

- a Testometric M500-100CT tensile-strength testing machine with steel and composite samples,
- a MIRA3 TESCAN analyzer for SEM analysis of the samples,
- a hardness tester for measuring the Vickers hardness HV1.

A macroscopic analysis of two welded joints was performed to assess the quality of the welded joints. Sample preparation was performed in a standard manner. It consisted of the sampling of a suitable size for embedding in the resin. Metallographic preparation consisted of embedding, grinding of the sample (grain size 800, 1200 and 2000) and polishing with diamond emulsions with gran sizes of 3 μm and 1μm. Etchant NaOH 1% was used for etching to induce structure.

## 3. Results and Discussion

Based on concept IV, we approached the technical solution of the fire-fighting adapter for the LKT, specifically the selection of adequate construction materials based on requirements for their strength and minimum weight. The basic condition for the design of the adapter was the use of the reserves of the capacity weight of the base machine with emphasis on:

- the center of gravity of the vehicle, depending on the change in driving characteristics,
- strength characteristics of the materials used for each part of the adapter,
- design and draft of the dimensions of the adapter, with respect to the dimensions of the attachment point,
- design of a water tank with a minimum water volume of 2000 L,
- design and draft of breakwaters in the tank,
- design of the protective frame of the tank.

The design of the fire-fighting adapter must be adapted to the parameters of LKT base machine. The LKT will be carried on the inclinable rear log arch. The fire-fighting adapter must also be able to be transported by other means, so it can be dismantled. The proposed fire-fighting body with its equipment will meet the criteria for forest fire interventions and the monitoring of forest areas.

The basic technical parameters for the fire-fighting adapter on LKT will be:

- water volume of about 2000 L,
- fire-fighting adapter equipment (high-pressure motor pump, fire extinguishing box),
- anchorage of the fire-fighting adapter on the inclinable rear log arch,
- the protective frame that ensures fire-fighting adapter handling (tank transfer, tank lifting),
- independent extinguishing after disconnecting from the base machine,
- the possibility of transportation of parts to build a lake system of water relay of forest fire-fighting.

Because the proposed fire-fighting body is to be used in the complicated terrain of mountain forests, its weight must be monitored in its design. Weight is a basic condition for ensuring good stability of the whole machine with an adapter. By reducing the weight, we could achieve better stability of the base machine. Stability, in the case of increased slope navigation or maneuvering on a slope could be increased by placing the adapter on the floor. In this case, it was necessary to use the floating position of the rear LKT shield. In order to increase the capacity weight of the base machine, but also to ensure maximum handling, a four-member operator was therefore applied for the lightweight construction of tanks of composite materials (fiberglass). [4] We came to these conclusions on the basis of an analysis of the mechanical properties of materials that could be used in the design of the tank.

On the Testometric M500-100CT tensile-strength testing machine we determined the basic strength properties (breaking strength $R_m$, yield point $R_p0.2$ and max. force at break of the sample $F_m$) and deformation characteristics (elongation A and reduction of area Z) of material samples of steel and fiberglass. We tested EN10028 P275N steel plates and fiberglass made of chopped strand mat M534 cured with polyester resin AROPOL M105 TB. Based on the measured values (Table 1), we can conclude that, in both materials, approximately the same maximum force $F_m$ required for tearing samples was achieved at break of the sample. The breaking strength $R_m$ of the steel was almost twice higher than that of the composite material. The yield point $R_p0.2$, which is characteristic of the plastic deformation zone, is identical for the composite to the achieved breaking strength $R_m$. Composites, such as amorphous materials, have low to zero plasticity. The steel samples had an $Rp0.2$ value higher than the composite material samples. The deformation characteristics of the composite sample cannot be determined for the reason already mentioned. A significant reduction of the area and elongation of the steel samples confirms the excellent plastic properties.

**Table 1.** Resulting values of the static tensile test.

| Test | $F_m$ [1] [N] | $R_m$ [2] [N/mm²] | $R_p0.2$ [3] [N/mm²] | $Z$ [4] [%] | $A$ [5] [%] |
|------|------|------|------|------|------|
| | **EN10028 P275N Steel** | | | | |
| 1 | 15,130 | 403,467 | 293,067 | 69,760 | 26,250 |
| 2 | 15,690 | 418,400 | 309,067 | 66,000 | 28,750 |
| 3 | 15,650 | 417,333 | 307,215 | 72,093 | 31,875 |
| | **fiberglass** | | | | |
| 1 | 16,360 | 218,133 | - | - | - |
| 2 | 14,340 | 191,200 | - | - | - |
| 3 | 16,210 | 216,133 | - | - | - |

[1] max. force at break of the sample; [2] breaking strength; [3] yield point; [4] reduction of area; [5] elongation.

The stress-strain diagram of the steel sample is shown in Figure 3a and the composite material is shown in Figure 3b.

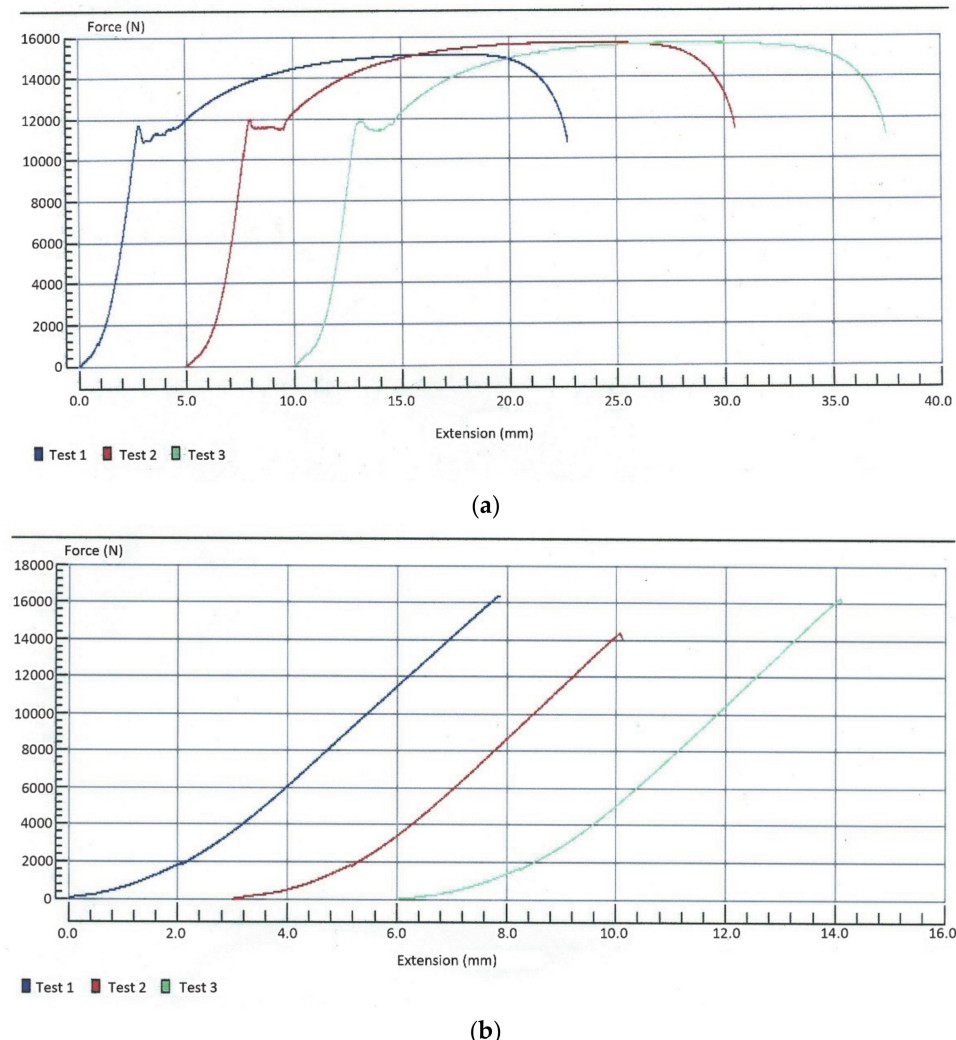

**Figure 3.** Stress-strain diagram: (**a**) steel sample; (**b**) composite material (fiberglass).

According to [4], when monitoring the weights, a significant difference is evident between the weight of the steel and fiberglass water tanks. The weight of fiberglass tanks is 82% lower compared to steel tanks. This fact and the analysis undertaken favor fiberglass for making these types of tanks.

According to [5], the type of resin used, the type of reinforcement, the proportional share of glass fibers in the total volume, and their arrangement give the properties of the fiberglass. Strength increases with increasing fiber content and their arrangement in one direction. Good interaction of the glass fibres with the matrix in the form of epoxy was proved on the base of the conclusions of mechanical testing and the subsequent results of the electron microscopy [6]. As early as in the design stage, the individual components were dimensioned to withstand many times greater loads and stresses than were envisaged for operational conditions [7]. Analysis was needed to assess the suitability of the material proposed by the tank in terms of its strength and possible material savings in its production. SEM analysis of the samples was performed at the Faculty of Engineering, Czech University of Life Sciences Prague by the MIRA3 TESCAN analyzer. The microstructure of the fibers of the composite used is shown in Figure 4. For this reason, the water tank does not exceed 130 kg. In the future we may also see the possibility of producing composite water tanks using natural fibers, to which various authors currently devote their research [8–16]. The replacement of synthetic fibers with natural ones has a lot of benefits which can be rationalized also by means of an ecological equilibrium [8].

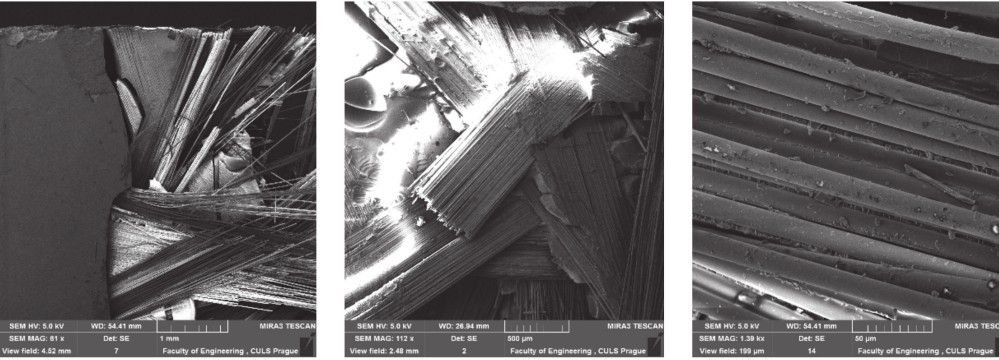

**Figure 4.** SEM images of the composite.

Good interaction of the glass fibres with the matrix in the form of epoxy was proved on the base of the conclusions of mechanical testing and the subsequent results of the electron microscopy [6].

As early as in the design stage, the individual components were dimensioned to withstand many times greater loads and stresses than were envisaged for operational conditions [7].

Replacing of the synthetic fibres with natural ones has a lot of benefits which can be rationalized also by means of an ecological equilibrium [8].

For reason of transport safety, and prevention of vertical and horizontal movement of water in the water tank during transport, as shown in Figure 5, we considered the construction of internal wave-breaks. These breakwaters are part of the construction of the water tank itself, thereby increasing its overall strength. To simplify this, maintenance has been undertaken to construct a removable lid.

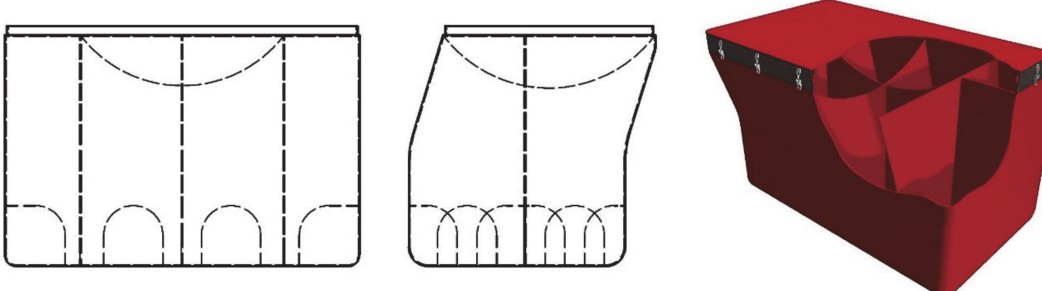

**Figure 5.** The water tank construction with built-in wave-breaks.

Requirements for the transportation of the tank in the forest terrain, and the possibility of its damage, required the construction of the protective frame (Figure 6), which performs a protective function during transport, but also when manipulating the water tank outside of transportation. In addition, in an emergency, it can serve as an improvised trailer for transporting materials in difficult terrain. In this case, the composite tank is removed from the protective steel frame.

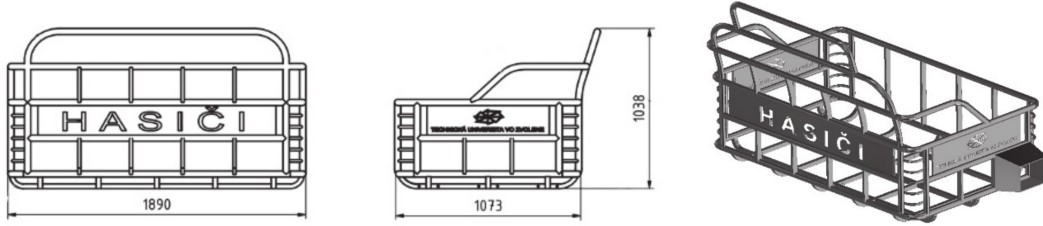

**Figure 6.** The protective frame construction of the tank for transporting water.

The frame is made of structural steel. In the future, the replacement of steel elements is being considered, for the purpose of lightening, with a solution being sought in another design material. Replacing steel elements with aluminum could contribute to lightening the weight of the entire adapter. This fact is based on the requirements of weight reduction in parts of the structure, where it can be done. The tank itself is maximally lightened (being constructed of fiberglass) and so it remains to solve this situation with the protective frame of the tank.

The possibility of replacing the steel welded structures with aluminum mainly require knowledge of the mechanical and technological properties of aluminum replacement, of which the weld ability is essential.

The welded joint usually determines the properties of the welded part. The structural differences of the welded joint from the structure of the base material affect not only the mechanical and fatigue properties of the joint, but also its corrosion resistance. Changes in the structure and thus the change in properties of a welded joint depend both on the welding method used and on the type of materials to be welded [17,18].

The welding of aluminum and its alloys has been understood for several decades. Nevertheless, aluminum has some specific properties that cause considerable problems during welding. Aluminum has a low melting point (approx. 660 °C) and its color does not change when it enters a liquid state. It has a very high thermal conductivity, which means that it needs a high concentration of energy for melting. Furthermore, it has a high affinity for oxygen and preferentially produces $Al_2O_3$. This oxide creates gases on the surface, causing it to act as a thermal and electrical insulation barrier, which makes welding impossible. Aluminum oxide $Al_2O_3$ is hygroscopic, which means there is a risk of hydrogen pores. This is due to the different solubility of hydrogen on the molten and cold metal. Aluminum is also sensitive to $CO_2$ when welding, which can also form pores even at low concentrations [19]. Faults and defects in welded joints are an unwanted but integral part of making welded joints. Therefore, the effort is to eliminate their occurrence as much as possible.

The use of the aluminum alloy AlSiMgMn EN AW-6082 has been suggested for some parts to achieve lightening of the adapter frame. This includes the corner parts of the structure, the protective cover of the outlet valve etc. This does not apply directly to the supporting or connecting elements to the adapter structure itself. Welding wire AlMg5Cr (5356) is recommended as a filler weld material. Due to its properties, it is the most widely used wire in the aluminum alloy category ([19,20]).

The welded joint (Figure 7a) has no visible gas pores, cavities, a lack of penetration and cracks. However, we can state that it has a surmounted arch weld. According to the standard described in [21], it is a defect from group five "Defects of shape and dimensions", with the numerical designation 503. The degree of weld quality for the design is defined by the mark D, that is, moderate level of quality. According to the standard described [22], the weld meets the condition of classification.

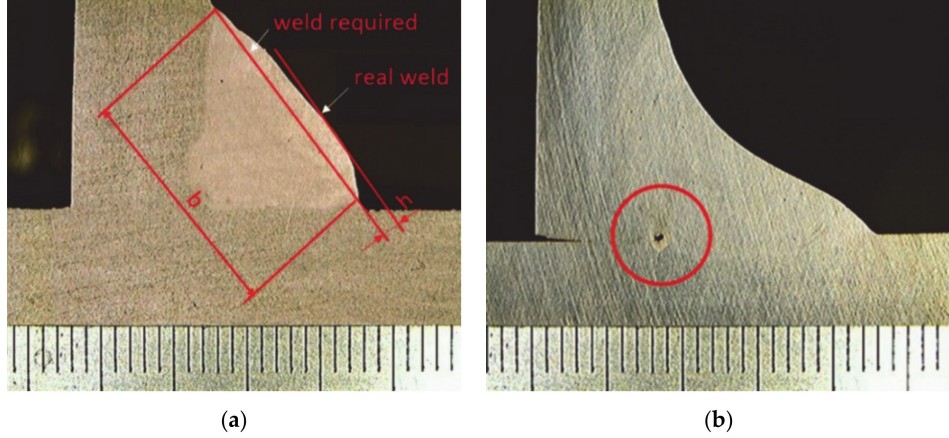

(**a**)                                        (**b**)

**Figure 7.** Macroscopic analysis of welded joints: (**a**) weld I; (**b**) weld II.

Figure 7b shows the gas cavity at the root of weld. The gas cavity is formed by trapped gases. It is a mainly spherical gas pore. According to the standard described in [21], it is a defect from group two "Cavities (gas pore)", with the numerical designation 2011. The degree of weld quality for the design is defined again by the mark D, that is, moderate level of quality. The level according to [22] satisfies the classification weld condition. According to the standard described in [22], the weld meets the condition of classification.

Furthermore, a Vickers HV1 hardness test was conducted [23,24]. The measurement was performed on sample II shown in Figure 8a. Figure 8b shows a graph of hardness progress from the base material through the weld and back to the base material.

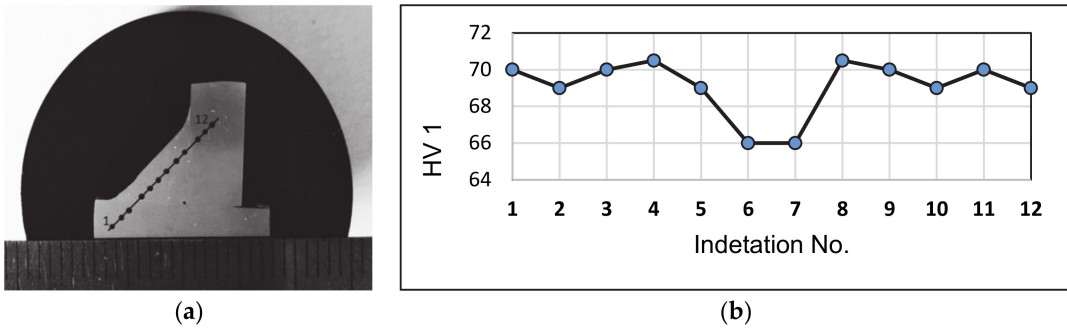

(**a**)　　　　　　　　　　　　　　　　　　　　　　(**b**)

**Figure 8.** HV1 measurement on sample: (**a**) weld sample II; (**b**) recording of hardness in weld joint.

Using the proposed methods for evaluating welds, we examined whether the welds had sufficient mixing of the base and filler material without visible defects such as cracks, seams or cold joints. Although the proposed welds are not directly bearing welds, according to the standard described in [21] which states that cracking is not acceptable in any case. The degree of quality of welds for the design is defined by the mark D, a moderate level of quality. According to the standard described in [22] both welds meet the classification condition.

Values from the base material as well as from the weld joint were obtained by evaluation of hardness by the HV1 measuring method. The graph (Figure 8b) shows that there was a decrease in hardness of about 5 Vickers units in the weld. On the other hand, the highest hardness (70HV1) was in the transition zone. Based on the results, we can conclude that the weld joint should be strong enough to withstand the load it is exposed to in operation.

It can be stated that the welds produced meet the basic requirements for this type of weld. The welding parameters were appropriately selected. After further testing and experiments, the proposed weld joints can be used in adapter frame construction.

Scheme supporting fire-fighting frame adapters were processed in Autodesk Inventor using the Shape Generator. This tool represents a new form of computer-aided structural design that provides capabilities beyond classical optimization. It is based on a generative design and simulates the natural evolutionary approach. Using a generic design does not result in one solution but a potentially large number [25]. The created structure is discontinuous and cannot fulfill its function in the generated state. The built-in algorithm detracts from the generated shape until completion of a specified criterion of weight loss. When replacing such a shape with a real construction, we will not directly arrive at an acceptable design (Figure 9).

To adjust the input form design, we therefore decided to reinforce the structure based on the results of regular stress analysis. We designed and applied several reinforcements (Figure 10) and performed a stress analysis (Figure 11). Due to the above-mentioned complex theoretical assumptions, we calculated a load of 50 kN, while the weight of the complete body, including the full tank is less than 2400 kg. This compensates for the fact that even if the device is not designed for dynamic driving, dynamic effects may occur.

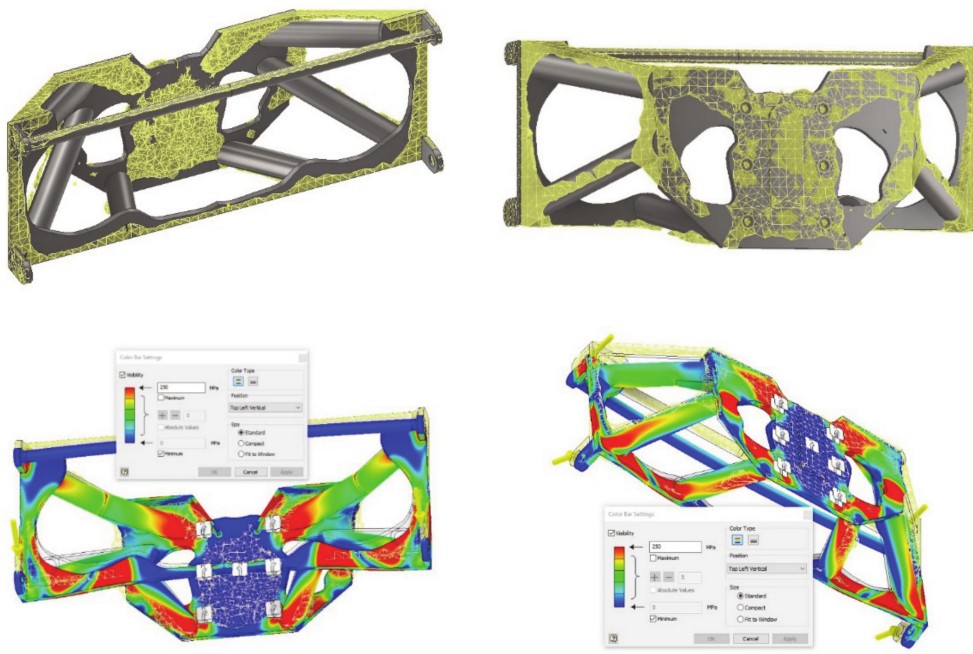

**Figure 9.** Replacement of the generated shape by a real construction—the maximum stress was significantly above the stress displayed.

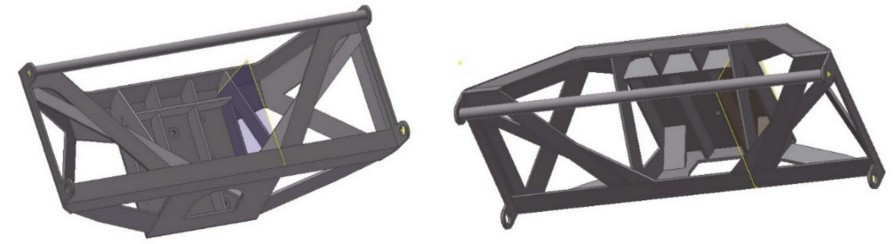

**Figure 10.** Modifying the supporting frame.

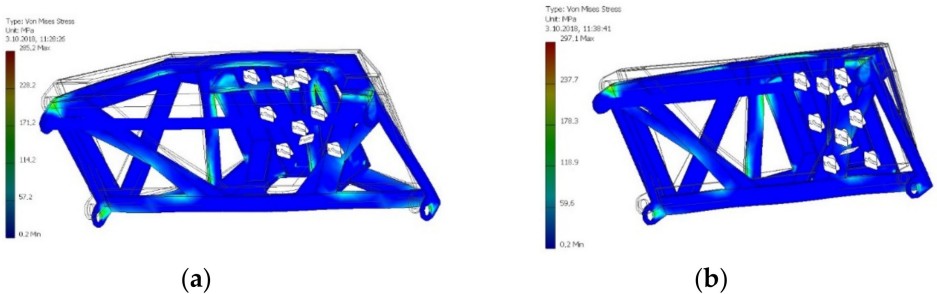

(**a**)                                                                           (**b**)

**Figure 11.** FEA analysis: (**a**) for slope gradient 0 %; (**b**) for slope gradient of 20%-driving uphill.

The basic essence of the technical solution of the fire-fighting adapter is to ensure sufficient water transport with the necessary firefighting equipment for firefighting purposes in conditions of mountain forests. The resulting technical solution of the fire-fighting adapter (Figure 12) is adapted to the parameters of the base machine LKT, on which the adapter is supported (Figure 13), respectively, semi-mounted (floating position) on the rear LKT shield. The adapter can be removed and transported by other means of transport to the place of intervention.

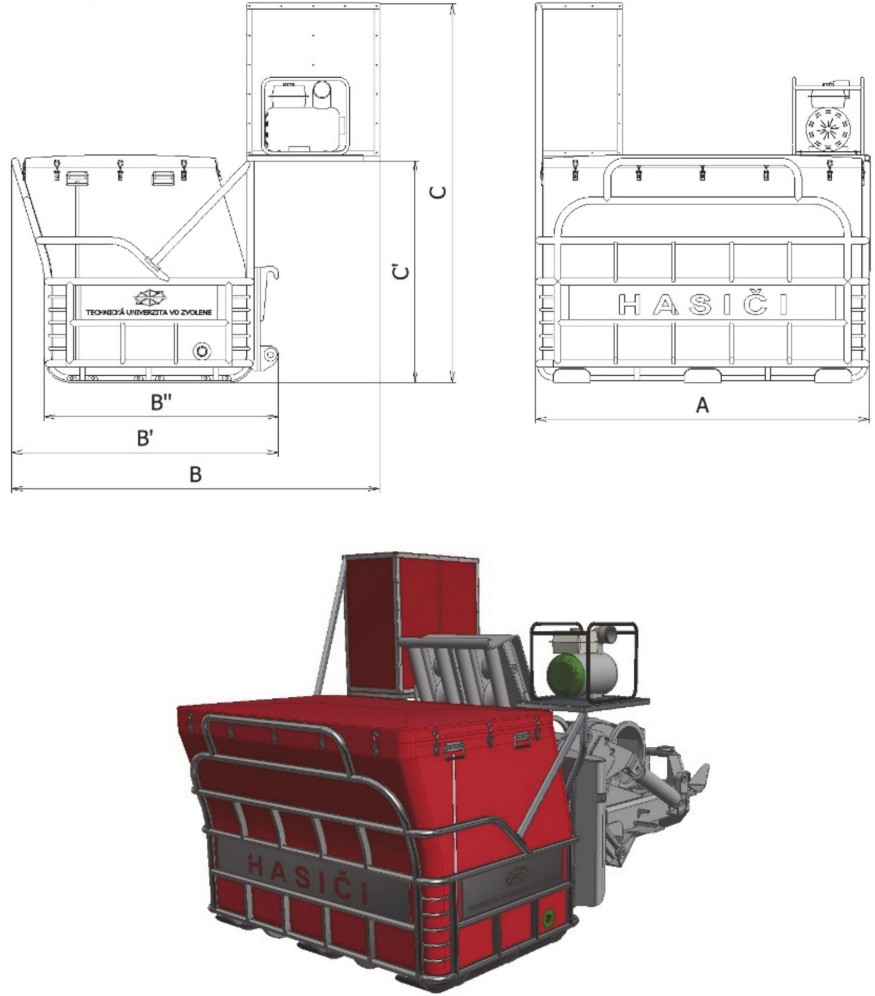

**Figure 12.** The design of the fire-fighting adapter. A = 1390 mm, B = 2082 mm, B′ = 1510 mm, B″ = 1325 mm, C = 2140 mm C′ = 1252 mm.

Additional equipment of the fire-fighting adapter according to the demands of accepted practice:

- water pump,
- 2″ suction hose (length 125 cm),
- 2″ suction hose with strainer (length 250 cm),
- hose "C" (fire hose with connector) 3 × 20 m,
- hose "D" (fire hose with connector) 4 × 20 m,
- dividing breeching C-DCD,
- hand branchpipe ("C" Profi, "D" Profi),
- ball valve "C" with connectors,
- the ax-hoe, the spade and the shovel.

This equipment is intended to ensure an autonomous capacity to refill the tank of the fire-fighting adapter from the nearest natural water source, as well as to manage the intervention itself in complicated terrain.

In addition to the use for fighting forest fires, the fire adapter can also be used to provide for transport of water for forest nurseries (irrigation), freshly planted areas in the event of prolonged drought, filling of watering-places for forest animals and filling puddles in the dry season.

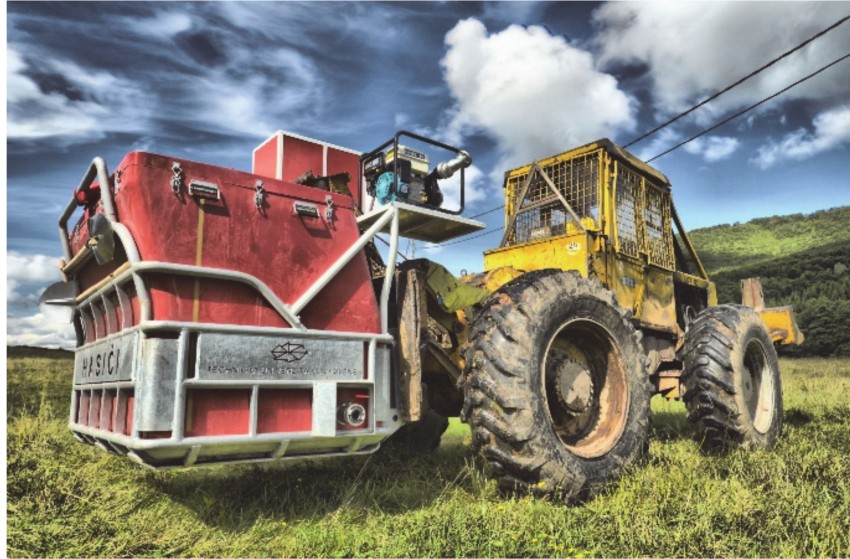

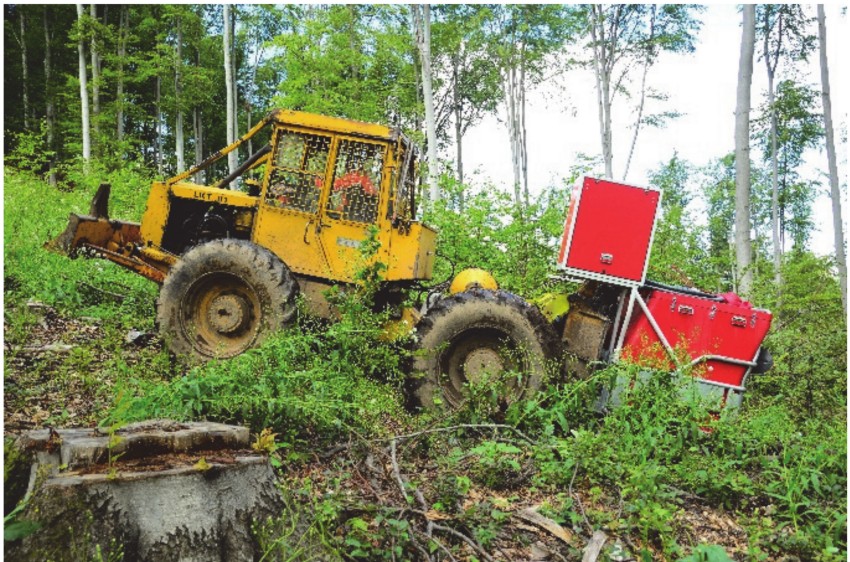

**Figure 13.** Fire-fighting adapter.

## 4. Conclusions

At present, the representation of base machines of the LKT type predominates in the forests of the Slovak Republic, but they do not have a similar extension, to the one introduced in the present article. Based on the above analysis, it would be appropriate to prepare a proposal on the basis of which a fire-fighting adapter of the said construction could be incorporated into the vehicle fleet. The adapter can be quickly mounted on an LKT base machine, which creates an effective tool in extinguishing fires, especially in inaccessible terrain.

The fire adapter (Figure 13); its technical design allows the transport of water into the area of fire and fire-fighting while connected to the LKT base machine. It has an autonomous drive (its own motor pump) for filling from a water source and to supply a hose on the offensive line. This ensures its full functionality even after disconnecting the LKT base machine.

Based on the field conditions of forests and the available technology, it can be stated that the proposed construction of the body is a new solution for the liquidation of forest fires in difficult

forest conditions. From the requirements of the fire brigade, as well as the forest management, one of the biggest problems in extinguishing forest fires is the transport and provision of a sufficient amount of water to the place of intervention. The most appropriate solution for such activities will be the designed fire-fighting adapter carried on a forest wheeled skidder (LKT). It is based on basic requirements such as maneuverability in a given space, sufficient power of the drive unit (base machine) and availability. Utility models: Protipožiarny adaptér na lesný kolesový ťahač/Fire-fighting adapter for forest wheeled skidder. Utility Model Number 8704, Publication of Registration Date 03.03.2020. Industrial Property Office of the Slovak Republic-WebRegisters https://wbr.indprop.gov.sk/WebRegistre/UzitkovyVzor/Detail/59-2019.

**Author Contributions:** Conceptualization, R.H., M.Ť. and M.H.; methodology, R.H.; validation, R.H., M.Ť., M.H. and V.M.; formal analysis, R.H.; investigation, R.H., M.Ť., M.H., J.M. and V.M.; resources, R.H., M.Ť. and M.H.; data curation, R.H., M.Ť., M.H. and J.M.; writing—original draft preparation, R.H.; writing—review and editing, R.H.; visualization, R.H., M.Ť., M.H. and J.M.; supervision, R.H. and M.Ť.; project administration, R.H. and V.M.; funding acquisition, R.H. and V.M. All authors have read and agreed to the published version of the manuscript.

**Funding:** This research was funded by "Agentúra na podporu výskumu a vývoja MŠVVaŠ SR" (contract No. APVV-14-0468 and No. APVV-16-0194).

**Acknowledgments:** This work was supported by "Agentúra na podporu výskumu a vývoja MŠVVaŠ SR" under the contract number APVV-14-0468 and partly by "Agentúra na podporu výskumu a vývoja MŠVVaŠ SR" under the grant number APVV-16-0194.

**Conflicts of Interest:** The authors declare no conflict of interest.

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
