# Peer review of "The Innovative Design of the Fire-Fighting Adapter for Forest Machinery"

_forests, doi:10.3390/f11080843_

Round 1

Reviewer 1 Report

The article is very interesting and valuable. It shows whole process of desinging of innovative fire-fighting attachment starting from a concept and ending with a product.

I recommend some small changes:

  1. Tab. 1. In a table or in a text. Add the steel title or EN steel number, add a type or a short description of tested fiberglass.
  2. Fig. 4 - Too small axis labels. Description "a" "b" in wrong places.
  3. Fig. 6 - bigger or clearer picture or close up for internal wave-breaks. Show at least one dimension on it.
  4. What about stability and power demand - are there any recommendations for a machine like weight?

Reviewer 2 Report

An interesting concept of an innovative adapter for a skidding tractor, the dominant model in the forestry of Central Europe. It allows to extend the possibilities of using the base machine for other tasks. Project work with great application potential; carried out in accordance with the current methods in this area.

Detailed comments:

(1) The manuscript does not contain abstracts and keywords.

(2) The authors analyze only one (IV) concept of the new construction, therefore the remaining possible should be criticized in the chapter discussion.

(3) Material testing methods should be in Chapter 2.

(4) The mechanical properties tests were carried out in accordance with the applicable standards, therefore Figure 3 does not add any new information and should be deleted.

(5) Rows 298-301 should be in Chapter 4.

(6) The authors should explain why they put so much work into material research, leading to a reduction in the weight of the adapter. They got a lot of information added. However, considering the large total weight of the container, was it crucial to reduce the weight of the structure itself?
